# Is ambient air pollution associated with onset of sudden infant death syndrome: a case-crossover study in the UK

Ian J Litchfield,[1] Jon G Ayres,[1] Jouni J K Jaakkola,[2] Nuredin I Mohammed[1,3]

[1]Institute of Applied Health Research, College of Medical and Dental Sciences, University of Birmingham, Birmingham, UK
[2]Center for Environmental and Respiratory Health Research, Faculty of Medicine, University of Oulu, Oulu, Finland
[3]Medical Research Council Unit The Gambia, Banjul, Gambia

**Correspondence to**
Dr Ian J Litchfield;
i.litchfield@bham.ac.uk

## ABSTRACT

**Objectives** Air pollution has been associated with increased mortality and morbidity in several studies with indications that its effect could be more severe in children. This study examined the relationship between short-term variations in criteria air pollutants and occurrence of sudden infant death syndrome (SIDS).

**Design** We used a case-crossover study design which is widely applied in air pollution studies and particularly useful for estimating the risk of a rare acute outcome associated with short-term exposure.

**Setting** The study used data from the West Midlands region in the UK.

**Participants** We obtained daily time series data on SIDS mortality (ICD-9: 798.0 or ICD-10: R95) for the period 1996–2006 with a total of 211 SIDS events.

**Primary outcome measures** Daily counts of SIDS events.

**Results** For an IQR increase in previous day pollutant concentration, the percentage increases (95% CI) in SIDS were 16 (6 to 27) for $PM_{10}$, 1 (−7 to 10) for $SO_2$, 5 (−4 to 14) for CO, −17 (−27 to −6) for $O_3$, 16 (2 to 31) for $NO_2$ and 2 (−3 to 8) for NO after controlling for average temperature and national holidays. $PM_{10}$ and $NO_2$ showed relatively consistent association which persisted across different lag structures and after adjusting for copollutants.

**Conclusions** The results indicated ambient air pollutants, particularly $PM_{10}$ and $NO_2$, may show an association with increased SIDS mortality. Thus, future studies are recommended to understand possible mechanistic explanations on the role of air pollution on SIDS incidence and the ways in which we might reduce pollution exposure among infants.

## INTRODUCTION

The quality of ambient air is an important factor in the health of adults and children. Ambient air quality is the second largest challenge facing public health in the UK.[1] According to WHO, over 3.7 million premature deaths per annum may be attributed to the harmful effects of ambient air. Children, it would seem, are more vulnerable than any other group,[2] and recent studies have indicated how even low levels of traffic-related air pollution can have a negative impact on birth outcomes and perinatal health.[3 4] A number of studies have also suggested a link between

---

**Strengths and limitations of this study**

► Sudden infant death is the leading cause of death in healthy infants between 1 month and 1 year old, and our study is the first based on time series data from the UK to investigate the relationship between common air pollutants and sudden infant death syndrome (SIDS).

► Using case-crossover methdology meant we were able to investigate various lags and multi-pollutant models and determine the persistent effects of single pollutants .

► Though our study is limited in power due to the comparatively small number of daily SIDS events, our chosen design is widely applied in air pollution studies and particularly useful for estimating the risk of a rare acute outcome associated with short-term exposure.

---

ambient air quality and the incidence of sudden infant death syndrome (SIDS).[5–12] Here, we examine the effects of the short-term variations in air pollution and the onset of SIDS.

Sudden infant death is the leading cause of death in healthy infants between 1 month and 1 year old,[13] and the impact on the families is notably traumatic as the death is without warning or witness.[14] The exact cause continues to tax researchers[15] though is likely the result of a combination of factors including susceptibility and environmental stressors such as lower social status of parents,[16] environmental tobacco smoke,[16 17] the prone position[18] and the winter season.[5 18] The impact of tobacco smoke suggests a respiratory trigger may be involved, however, the evidence of an association between SIDS and air pollution warrant further research as findings from epidemiological studies have been inconsistent[19 20] and few have satisfactorily explored the impact of short-term exposure on SIDS. Our study is the first based on time series data from the UK to investigate the relationship between common air pollutants and SIDS. We have collated data from a 10-year

period on concentrations of air pollution and onset of SIDS within the West Midlands one of the largest and most polluted conurbations in the UK and have conducted a case-crossover study to determine any associations.

## METHODS
### Settings
The West Midlands is a metropolitan county in the centre of the UK. It has a population of some 2.8 million.[21] The West Midlands is one of the most heavily urbanised counties in the UK and forms the most populated conurbation in the UK outside London, it is at the heart of the UK motorway network and remains a significant centre of the UK's manufacturing industry.

### Data collection
We combined data on SIDS events with data on total births, air pollution, air temperature and a measure of deprivation.

### Data on SIDS mortality and total births
We obtained daily time series data on SIDS mortality (International Classification of Diseases(ICD)-9: 798.0 or ICD-10: R95) for the period 1996–2006 from the Perinatal Institute. All cases were between 0 and 12 months old at onset. This data consisted of the date of death and the first three digits of the postal code to avoid the possibility of identifying individual cases due to the rare nature of SIDS. This allowed us to explore the effects of short-term exposure. We obtained daily births data with West Midlands' postal codes from the Office of National Statistics for the period 1996–2006 and was used for descriptive analysis only.

### Air pollution data
The daily time series data on air pollution were compiled from the UK air quality archive managed by Department for the Environment, Food and Rural Affairs. These include a total of 10 monitoring stations in the West Midlands measuring $PM_{10}$, $SO_2$, $NO_2$, NO, NOx, CO and $O_3$ including sites within Birmingham, Coventry, Walsall

and Wolverhampton. Before 1996, the reliability of air quality data was inconsistent. Not all monitoring stations had measurements for the range of pollutants over the entire study period and the precise pollutants measured at each monitoring centre are described in table 1.

We aimed to examine the association between day-to-day variability in air pollution exposure and SIDS counts over the entire region rather than contrasting exposure and outcome between areas within the West Midlands region. Therefore, the daily pollution data for West Midlands were calculated by averaging across all monitoring stations with available measurements as our models were based on the temporal relationship between air pollution and SIDS; we did not fit a spatial model.

### Temperature data
Data on daily minimum and maximum temperature were compiled for weather monitoring stations in the West Midlands from the Meteorological Office British Atmospheric Data Centre.[22] We used the average daily temperature which was calculated by taking the average of the minimum and maximum temperature at each monitoring station for each day to obtain the daily average temperature for the entire West Midlands.

### Index of Multiple Deprivation score
The Index of Multiple Deprivation (IMD) score is a composite measure based on seven dimensions of deprivation, including income deprivation, employment deprivation, health deprivation and disability, education deprivation, crime deprivation, barriers to housing and services deprivation and living environment deprivation. The data for the 2010 IMD at postal code level were downloaded from EDINA Digimap ShareGeo facility, an online spatial data repository and was used for descriptive analysis only.[23]

### Statistical analysis
The case-crossover design was used to investigate the association between short-term exposure to air pollution and

**Table 1** Pollutants measured and corresponding time period by monitoring stations

| Name (postal code area) | Pollutants | Time period* |
|---|---|---|
| Birmingham Centre (B) | $PM_{10}$, $SO_2$, $NO_2$, NO, NOx, CO, $O_3$ | 01/01/1996 to 31/12/2006 |
| Birmingham East (B) | $PM_{10}$, $SO_2$, $NO_2$, NO, CO, $O_3$ | 01/01/1996 to 03/08/2004 |
| Birmingham Tyburn (B) | $PM_{10}$, $SO_2$, $NO_2$, NO, CO, $O_3$ | 16/08/2004 to 31/12/2006 |
| Oldbury (B) | $PM_{10}$, $SO_2$, $NO_2$, NO, NOx, CO, $O_3$ | 27/06/1997 to 20/09/1998 |
| West Bromwich (B) | $PM_{10}$, $SO_2$, $NO_2$, NO, NOx, CO, $O_3$ | 04/11/1998 to 31/12/2006 |
| Coventry Centre (CV) | $PM_{10}$, $SO_2$, $NO_2$, NO, NOx, CO, $O_3$ | 19/02/1997 to 31/12/2006 |
| Coventry Memorial (CV) | $PM_{10}$, $SO_2$, $NO_2$, NO, NOx, CO, $O_3$ | 26/02/2001 to 31/12/2006 |
| Walsall Alumwell (WS) | $NO_2$, NOx | 01/01/1996 to 31/12/2006 |
| Walsall Willenhall (WS) | $NO_2$ | 13/05/1997 to 31/12/2006 |
| Wolverhampton Centre (WV) | $PM_{10}$, $SO_2$, $NO_2$, NO, NOx, CO, $O_3$ | 19/12/1995 to 31/12/2006 |

*Period with at least one pollutant being measured and taking into account missing data between start and end dates.

the occurrence of SIDS events controlling for average daily temperature and national holidays. This design, introduced by Maclure,[20] has been widely applied in air pollution studies and is particularly useful for estimating the risk of a rare acute outcome associated with short-term exposure.[24–27] In case-crossover design, each case acts as their own control and like case–control studies,[28] the distribution of exposure is compared between 'cases' and 'controls'. That is, exposure at the time just prior to the event ('case' or 'index' time) is compared with a set of 'control' times that represent the expected distribution of exposure for non-event follow-up times. The design helps primarily to control for confounding by subject-specific factors which remain static over time such as ethnicity and gender.

We applied the time stratified case-crossover approach which has previously been used to minimise bias.[29] That is, control days were selected from the same day of the week, within the same calendar month and year as the event day. We applied a conditional Poisson regression model which has been shown to give equivalent estimates as the conditional logistic model but with the advantage of readily allowing for overdispersion and autocorrelation.[30] All our models assume a linear effect for air pollution as reported in previous studies[31–33] while temperature effects are likely to be non-linear and were modelled using natural splines with 3 df.[34–36]

### Sensitivity analyses

The primary aim was to investigate the risk of SIDS events in relation to air pollution on the previous day in single-pollutant models. To examine sensitivities to our a priori model specification, additional lag structures were explored including single lags of 0, 1, 2,…, 6 days and also the corresponding average of lags 0–1, 0–2, 0–3,…, 0–6. Moreover, the association between SIDS and air pollution was examined after adjusting for the effect of each of the pollutants $PM_{10}$, $SO_2$, $NO_2$, NO, CO and $O_3$ as a second pollutant in turn. The aim of this two-pollutant modelling approach was to account for potential confounding effect of copollutants. Further sensitivity analyses were performed by controlling for minimum temperature instead of the average temperature. We excluded NOx from the sensitivity analyses as it showed very strong correlation particularly with NO (r=0.96).

Results are presented as percentage increases in mortality with 95% CIs for an IQR increase in pollutant concentration. Hypothesis tests were two sided with a significance level of 0.05. All analyses were performed using the R statistical software; details of the packages used and sample line of code for fitting specific conditional Poisson regression model is given in the supplementary material (see online supplementary file S1).[37]

## RESULTS
### Descriptive statistics
Over the study period (1996–2006), there were 211 SIDS events across the four postal code areas (B, CV, WS and WV) included in the analyses which accounted for about approximately 5% of the days within the study period (table 2). In the same period and location, a total of 943 937 live (single) births were registered.

The daily average air pollution concentrations and their SDs are presented in table 3 and show that average concentrations tended to be below UK air quality limits as defined by the EU Ambient Air Quality Directive.[38]

Table 4 shows that there was generally a strong correlation between the levels of pollutants investigated with the exception of the weaker correlation between $O_3$ and $PM_{10}$ (r=−0.26) and with $SO_2$ (r=−0.34). What was also notable was the negative correlation between ozone and the other pollutants and how in contrast to the other pollutants its positive correlation with temperature.

Comparing the four postal code areas, Birmingham had the highest SIDS mortality (about 56%) and births rates (about 80%). Air pollution concentrations were

**Table 2** Average air pollution (μg/m³), temperature (°C) and IMD scores for selected West Midlands postal code areas with SIDS and birth counts, 1996–2006

| Post town (area) | $PM_{10}$ | $SO_2$ | CO | $O_3$ | $NO_2$ | NO | NOx | Temperature (minimum-maximum) | IMD score | SIDS event (%) | Birth count (%) |
|---|---|---|---|---|---|---|---|---|---|---|---|
| Birmingham (B) | 23.8 | 6.8 | 0.4 | 38.5 | 33.9 | 17.5 | 60 | 7.2–12.8 | 36.6 | 118 (55.9) | 753 844 (79.9) |
| Coventry (CV) | 18.3 | 11.3 | 0.3 | 32.4 | 22.5 | 9.2 | 36.2 | 7.7–13.5 | 23.7 | 43 (20.4) | 91 393 (9.7) |
| Walsall (WS) | | | | | 41.5 | | 89.4 | 6.4–14.0 | 25.2 | 24 (11.4) | 53 532 (5.7) |
| Wolverhampton (WV) | 23.8 | 9.3 | 0.5 | 38.9 | 29.5 | 18.9 | 59.5 | 8.4–12.3 | 33.4 | 26 (12.3) | 45 168 (4.8) |
| Total | | | | | | | | | | 211 (100) | 943 937 (100) |

IMD, Index of Multiple Deprivation; SIDS, sudden infant death syndrome.

**Table 3** Descriptive statistics for daily SIDS mortality counts, air pollution (µg/m$^3$), temperature (°C) and birth counts for West Midlands, 1996–2006

| Variable | Mean | SD | Median | IQR | Minimum | Maximum |
|---|---|---|---|---|---|---|
| SIDS count | <1 | <1 | 0 | 0–0 | 0 | 2 |
| Average temperature | 10.2 | 5.4 | 10.2 | 6.2–14.5 | −4.2 | 25.0 |
| Maximum temperature | 13.1 | 6.0 | 12.9 | 8.5–17.6 | −2.1 | 29.6 |
| Minimum temperature | 7.4 | 5.1 | 7.5 | 3.6–11.4 | −7.5 | 20.7 |
| $PM_{10}$ | 23.4 | 11.6 | 20.2 | 15.7–28.2 | 4.0 | 128.9 |
| $SO_2$ | 8.5 | 6.9 | 7.2 | 4.4–10.2 | 0.0 | 70.8 |
| CO | 0.4 | 0.2 | 0.4 | 0.3–0.5 | 0.1 | 3.5 |
| $O_3$ | 38.1 | 18.2 | 38.0 | 25.4–50.4 | 1.3 | 115.8 |
| $NO_2$ | 33.2 | 14.3 | 31.2 | 22.3–41.8 | 5.9 | 113.0 |
| NO | 16.9 | 23.8 | 9.0 | 5.4–17.9 | 0.4 | 314.9 |
| NOx | 64.5 | 49.3 | 51.1 | 35.2–76.4 | 7.9 | 569.2 |
| Birth count* | 290 | 34 | 296 | 266–314 | 183 | 387 |
| IMD score | 32.3 | 12.6 | 31.7 | 21.3–40.5 | 7.2 | 54.9 |

*Counts are for single births only.
IMD, Index of Multiple Deprivation; SIDS, sudden infant death syndrome.

more or less similar for Birmingham and Wolverhampton except slightly lower levels of $NO_2$ and higher levels of $SO_2$ observed for the latter. Coventry had the lowest pollution level with respect to all pollutants except $SO_2$. Walsall had measurements for $NO_2$ and NOx only which were very large compared with the other postal code areas (table 2). Similarly, Birmingham and Wolverhampton had higher average deprivation scores compared with Coventry and Walsall (table 2). Also, SIDS counts and pollution levels tended to be larger in highly deprived areas and near motorways and A-road networks which may be a reflection of population density (figure 1; see online supplementary figure S2). There was also indication of seasonal pattern for SIDS occurrence; the highest proportions of SIDS were observed in January and February (about 10%) and among the relatively colder months and in July (about 14%) among the warmer months (data not shown).

### Case-crossover analysis
Figure 2 shows the estimated OR (95% CI) for the association of SIDS events with each of the pollutants considered based on the conditional Poisson model. For an IQR increase in previous day pollutant concentration, the percentage increases (95% CI) for the risk of SIDS death was 16 (6 to 27); p=0.002 for $PM_{10}$, 1 (−7 to 10); p=0.83 for $SO_2$, 5 (−4 to 14); p=0.33 for CO, −17 (−27 to −6); p=0.004 for $O_3$, 16 (2 to 31); p=0.02 for $NO_2$, 2 (−3 to 8); p=0.47 for NO and 8 (−1 to 18); p=0.07 for NOx after controlling for average temperature and national holidays. Therefore, considering pollutant levels at lag 1, significant association with increased risk of SIDS was indicated only for $PM_{10}$ and $NO_2$ while a protective effect was apparent for $O_3$.

### Sensitivity analyses
Lag structures: Analyses exploring relationships between SIDS and air pollution for other lag structures (lags 0, 2, 3,…, 6 and corresponding average of lags 0–1, 0–2, 0–3,…, 0–6) indicated associations were somewhat sensitive to the choice of lag (figure 3). In the single-lag models, there was an impression of stronger delayed effects for CO, $NO_2$, $O_3$, $PM_{10}$ and $SO_2$ compared with relatively recent lags (lag 0 and lag 1). The effect of NO appeared to remain comparatively flat across lags. Relatively more consistent association across lags was observed for $NO_2$

**Table 4** Pairwise Pearson correlations coefficients (r) for pollutants and temperature

| | $PM_{10}$ | $SO_2$ | CO | $O_3$ | $NO_2$ | NO | NOx |
|---|---|---|---|---|---|---|---|
| $SO_2$ | 0.53 | | | | | | |
| CO | 0.56 | 0.59 | | | | | |
| $O_3$ | −0.26 | −0.34 | −0.56 | | | | |
| $NO_2$ | 0.66 | 0.56 | 0.76 | −0.62 | | | |
| NO | 0.54 | 0.48 | 0.87 | −0.58 | 0.73 | | |
| NOx | 0.59 | 0.56 | 0.90 | −0.62 | 0.86 | 0.96 | |
| Temperature | −0.08 | −0.24 | −0.39 | 0.40 | −0.40 | −0.38 | −0.40 |

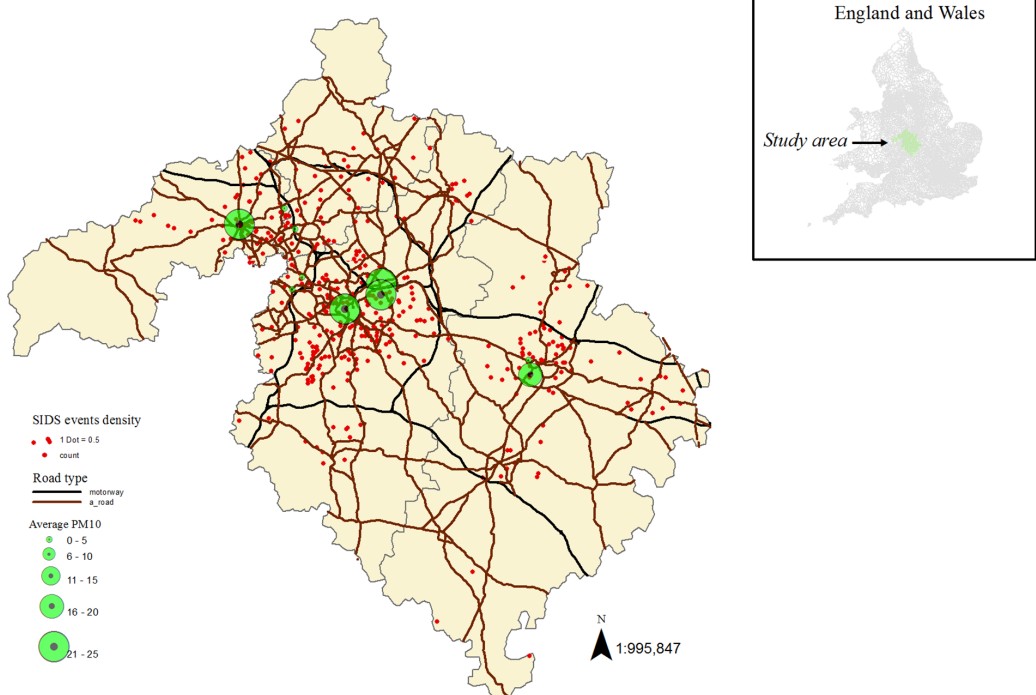

**Figure 1** Air pollution monitoring sites with road networks (motorways and A-roads) and sudden infant death syndrome (SIDS) events in the study area. We used ArcGIS Desktop V.10.2 (http://www.esri.com/software/arcgis) to create the map combining shapefiles for postal code areas and motorways and overlaying air pollution and SIDS data.

and $PM_{10}$ and to some extent for CO. Likewise, in average lag models, ORs tended to increase with averaging over more lags with exception of NO where such an effect was not observed. Results were also similar (at least qualitatively) after adjusting for minimum temperature instead of the average temperature (see online supplementary figure S3).

Multipollutant models: Further investigation using two-pollutant models also showed associations were sensitive to control of other pollutants. In general, adjusting for copollutants appeared to attenuate ORs towards the null except for $NO_2$ and $PM_{10}$ where effects remained to persist (figure 4). Interestingly, after controlling for

$PM_{10}$, the protective effect of $O_3$ and the adverse effect of $NO_2$ observed in the single-pollutant models were not apparent for most of the lag choices. Similarly, none of the other pollutants showed marked association with SIDS after controlling for $PM_{10}$ effects. The estimates from the multipollutant model should, however, be interpreted with caution as most of the pollutants were correlated.

## COMMENTS
### Summary of findings

In this study, we hypothesised a delayed effect (lag 1) of air pollution on SIDS incidence and investigated additional lags in sensitivity analyses and in single-pollutant and multipollutant models. Though CIs were wide, we found evidence suggesting association of SIDS mortality with $PM_{10}$ and $NO_2$ exposure. Compared with other pollutants, their effects persisted after controlling for copollutants and across the various lag structures investigated. The exception was controlling for $O_3$ did attenuate the risk estimates observed for both $PM_{10}$ and $NO_2$. There were no consistent associations observed with exposure to the remaining pollutants investigated (CO, $SO_2$, NO, NOx). An exception was the protective effect observed in relation to $O_3$ exposure.

### Strengths and limitations

Previous studies have tended to focus on relatively longer-term exposure to air pollutants.[37 39 40] Our study is one of the first to use case-crossover methodology to explore the impact of air pollution on SIDS based on data from UK.

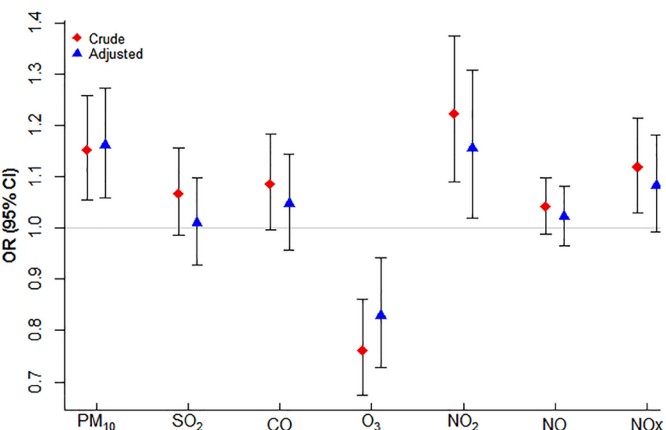

**Figure 2** Estimated risk of sudden infant death syndrome for an IQR increase in lag 1 pollutant concentration before and after controlling for confounding by average temperature and national holidays.

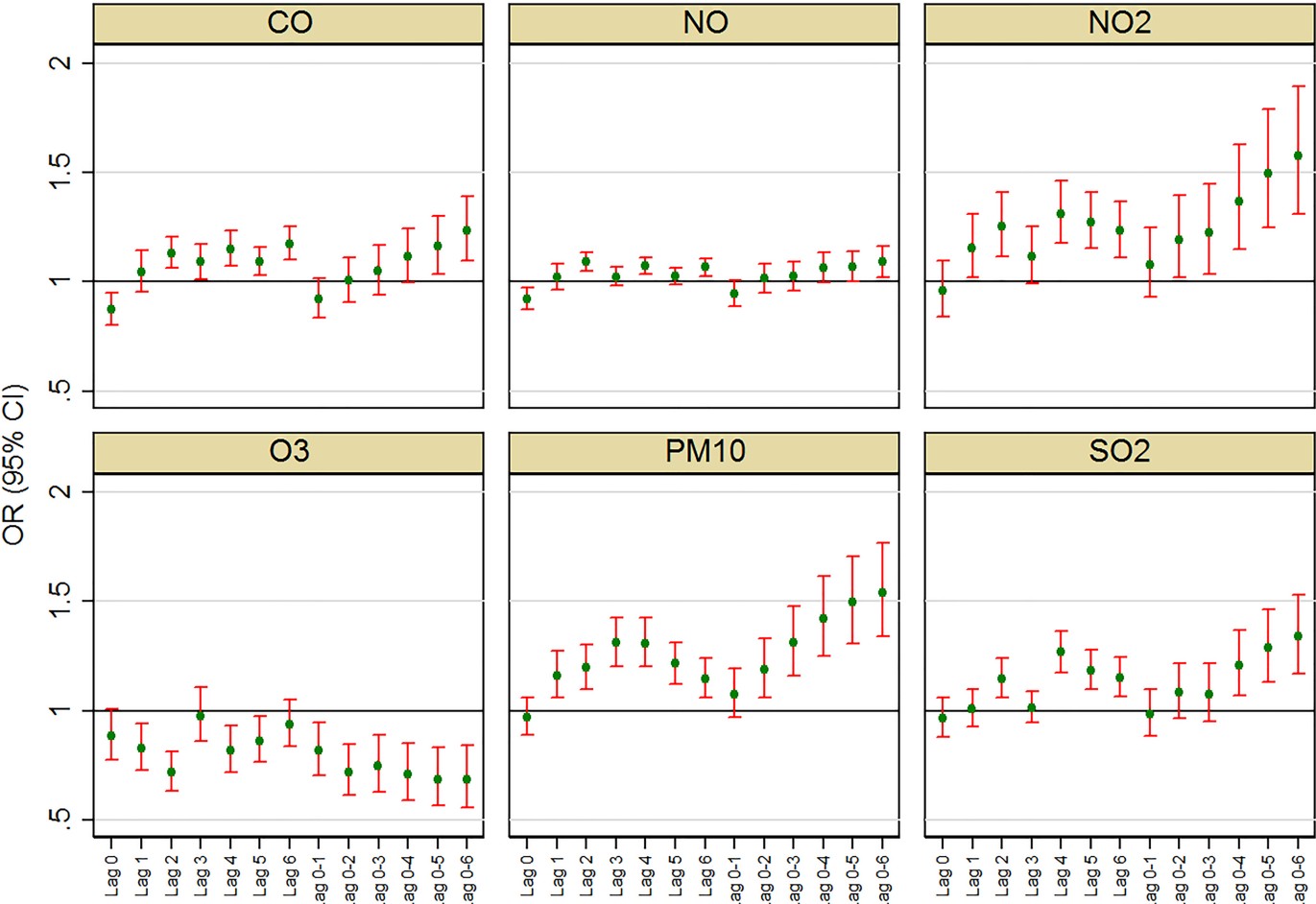

**Figure 3** Risk of sudden infant death syndrome associated with an IQR increase in pollution for selected lags and average of lags; models adjusted for average temperature and national holidays.

Using an approach that is suitable for estimating the risk of a rare acute outcome such as SIDS events, we were able to investigate various lags and multipollutant models indicating delayed effects of pollutants that persisted after controlling for copollutants. However, measurement error for air pollution exposure a potential limitation in this study as we did not use personal measurements. Other epidemiological studies using data from ambient monitoring stations have shown a good level of correlation between daily personal exposure and daily ambient air pollution measurements,[36 37] though there is the potential for measurement error that can attenuate regression coefficient estimates. That is, such misclassification will likely lead to an underestimation of the effects. We were also unable to control for influenza and humidity as we did not have access to reliable data for these variables.

### DETAILED DISCUSSION

Certain groups of the population are more vulnerable to ambient air pollution than others and children figure predominantly among them due to the fragility of their immune system and the ratio of their lung capacity to their size.[41] Particulate matter has been recognised as the most significant contributor to air quality-related morbidity and mortality[42] and a number of studies have reported a link between infant mortality and $PM_{10}$ at similar concentrations of pollutant below air quality guidelines as are described in this study.[11 43 44] However, previous evidence of a direct association with SIDS is inconclusive and few studies describe a significant association of $PM_{10}$ with increased risk.[6 45]

The potential association with $NO_2$ we discovered reflected the findings of a multicity study from Canada that found an increased risk for SIDS with a lag of 2 days.[8] More broadly, our findings are in line with previous studies that have found evidence of a lag between exposure to criteria pollutants and mortality. For $PM_{10}$, a single-day exposure has been shown to have an effect for up to 5 days[46 47] and numbers of respiratory deaths appear to be more affected by air pollution levels on previous days, than cardiovascular deaths that are impacted by same-day pollution.[48] For example, myocardial infarction represents an acute response to a trigger.[49]

This is biologically plausible when considering how a wider and more lagged response can be expected for deaths from respiratory responses to pollution (eg, chronic obstructive pulmonary disease) via the proposed

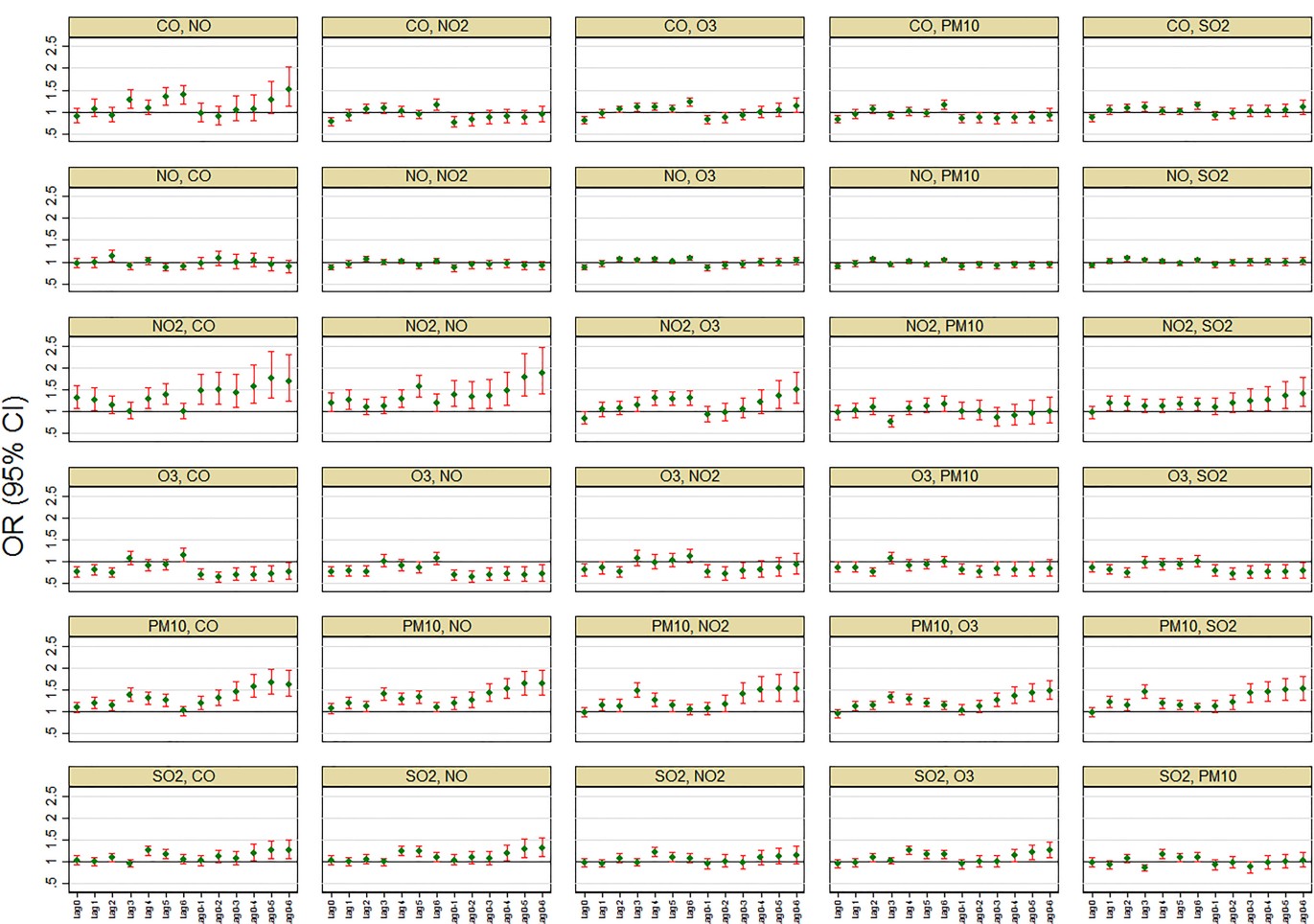

**Figure 4** Risk of sudden infant death syndrome associated with an IQR increase in air pollution after controlling for copollutants, average temperature and national holidays (panel titles indicate: main and copollutant, respectively).

mechanisms, such as pulmonary or systemic inflammation and modulated immunity.[50]

With regard to the apparent protective properties of $O_3$ that we describe here, its worth noting that these have been seen previously.[39] One possible reason for the negative association of $O_3$ and SIDS might be its negative correlation with other pollutants, in particular that with fine particulate matter pollution.[51] $O_3$ also showed a positive correlation with temperature while all other pollutants were negatively correlated.

Concern around the effects of air pollution continues to mount particularly within developing economies[52] where it contributes to 3.3 million premature deaths worldwide per year, a figure estimated to double by 2050 if the issue remains unattended.[53] However, it appears that recently fossil fuel emissions have begun to increase again due to the developing world's reliance on them to power their expanding economies.[54] In the developed West, policy initiatives are beginning to recognise the mounting issue posed by the adverse effects on health posed by ambient air pollution. However, shifting political priorities in the USA has seen a commitment to revitalise the coal industry[55] and an increase in the production of shale oil.[56] However, mitigating these risks is not a straightforward proposition and government policies appear slow to react, for example, in the UK pledges to cease sales of diesel and petrol cars do not come into effect until 2040.[57]

In the absence of coherent policy to address the issue, a number of proposals have been put forward to mitigate the effects of high levels of ambient exposure. Knowing that children and young adults may be highly susceptible to some of the subclinical changes caused by air pollution[58 59] and as indoor concentrations are lower than ambient levels, advice has been to remain indoors to reduce exposure and acute health risks on high air pollution days.[60] There are also systems available for cleaning indoor air though these may be deemed expensive for the economically deprived.[61] There has also been a case made for chemopreventive interventions, such as antioxidant or antithrombotic agents, but without data on health outcomes, no recommendations can be made in their use for primary prevention.[62]

## CONCLUSIONS

Understanding the effects of air pollution on child heath is more relevant than ever. Our work here has highlighted

a potential association of sudden infant death with $PM_{10}$ and $NO_2$ and the association with particulate matter and infant mortality in particular is widely recognised. However, until policy reflects the growing evidence and responds to mounting public concern, it would appear to be the responsibility of individuals to take independent action to mitigate the effects of air pollution and protect the health of their young ones.

**Contributors** IJL contributed to the conceptualisation of the study, initial drafting of the manuscript and supervised the research project at all stages. NIM contributed to the research design, performed the data management, analysis and interpretation, and with IJL drafted the initial manuscript. JGA contributed to the conceptualisation of the study, data interpretation and critically reviewed the manuscript. JJKJ contributed to the conceptualisation of the study, data interpretation and critically reviewed the manuscript. All authors approved the final manuscript as submitted.

**Funding** This work was supported by The Lullaby Trust, grant number 260.

**Competing interests** None declared.

**Patient consent** Not required.

**Ethics approval** Ethical approval was given by the University of Birmingham, Life and Health Sciences Ethical Review Committee.

**Provenance and peer review** Not commissioned; externally peer reviewed.

**Data sharing statement** No additional data are available.

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
