## [Reviewer comments · BMJ Open]

ARTICLE DETAILS

TITLE (PROVISIONAL)	Is Ambient Air Pollution associated with onset of Sudden Infant Death Syndrome: A case crossover study in the UK
AUTHORS	Litchfield, Ian; Ayres, Jon; Jaakkola, Jouni; Mohammed, Nuredin

VERSION 1 – REVIEW

REVIEWER	Takashi Yorifuji Okayama University, Japan
REVIEW RETURNED	12-Jul-2017

GENERAL COMMENTS	The present study examined the association between short-term variations in criteria air pollutants and occurrence of SIDS using data from the West Midlands region in the UK. The study is well-conducted and the manuscript is well-written. I have several comments for clarification. 1. The age range of the participants is not shown. I guess most of the participants are infants.2. The authors included SIDS mortality for relatively a long period (from 1996 to 2006). I wonder whether the definition of the SIDS has changed during the period. For example, metabolic diseases may have been included in the category of SIDS previously.3. It is not slightly clear from the manuscript how the authors assigned air pollution exposure, temperature, or other potential confounders (daily birth counts and IMD) to the participants. Do they only consider temporal variability or do they also consider geographical variability using postcodes?4. The authors adjusted for daily birth counts and IMD in the model, but I am a little skeptical about this. Because the authors used case-crossover study design, individual unchangeable characteristics like IMD are already adjusted in the design. Moreover, variations in daily birth counts would not affect occurrence of the SIDS and would not be associated with air pollution, thus daily birth counts does not fulfill the definition of confounding.5. Rather, some air pollution studies adjust for daily humidity, national holidays, or influenza occurrence which can change daily. These variables should be considered as other potential confounders.6. Discussion is too weak. Some discussions should be added. For
--

	example, why did the authors find the positive associations in lag 1 or later lags rather than the lag 0? What is the mechanism of the association between air pollution and SIDS? Other limitations such as potential outcome misclassification or residual confounding should be discussed.
--	---

REVIEWER	Dr. Shereen Hamadneh Al-albays University, Jordan
REVIEW RETURNED	18-Jul-2017

GENERAL COMMENTS	this study would be interested for the pediatric health research. would you include more details of your method, described sufficiently to allow the study to be repeated. this study aimed to examine the relationship between short-term variations in criteria air pollutants and occurrence of SIDS, would you please include research question or study objective and clearly defined in the introduction section under the problem statement. Good luck
--

REVIEWER	Hans Scheers currently: P95 Koning Leopold III laan 1 3001 Heverlee Belgium formerly: KU Leuven Department of Public Health Herestraat 49 3000 Leuven Belgium
REVIEW RETURNED	27-Jul-2017

GENERAL COMMENTS	Comments for authors In this paper, the authors investigated the association between Sudden Infant Death Syndrome (SIDS) and atmospheric air pollution in a densely populated region in the United Kingdom. They authors applied a case-crossover (CCO) design, a widely applied method to study acute health effects in association with short-term changes in air pollution. The authors found a significantly higher risk for SIDS with increasing levels of PM₁₀ and NO₂. For other pollutants, however, no such associations were detected, and O₃ had a protective effect. The authors conclude that air pollution exposure may increase risk of SIDS mortality.
--

The paper attracts attention by its conciseness, fluent language and clear message. However, the brevity of the manuscript has some drawbacks. Possible questions raised by the results remain unanswered in the discussion, and I also have a couple of questions on the methodology.

My detailed comments are as follows:

Major comments

My major comment concerns the brevity of the “Comments” section and subsequent lack of in-depth discussion. I see at least three points that deserve more attention:

1. Finding stronger associations with PM₁₀ and NO₂ than with other pollutants is in line with many other studies that included several pollutants. Particularly PM is considered the most harmful of all pollutants. You may want to mention this accordance and refer to some studies.
2. Finding stronger delayed effects (lags 3 to 6) than immediate effects (lags 0,1,2), however, is not in line with many CCO studies on air pollution. Please discuss probable mechanisms that might explain a delayed effect for SIDS mortality, in contrast to the immediate effects usually found when studying other outcomes, such as myocardial infarct or overall mortality. Also, the figure presenting results adjusted for IMD (Figure S2) shows remarkably high ORs for lag 6, compared to the other lag structures for most pollutants. Are you sure this figure is correct? It is hard to believe that adjusting for IMD leads to such a different figure than the one showing crude associations (Figure 2).
3. Next to biological mechanisms, I also would like to read some more discussion on the public health impact of your study. Just writing “Thus, future studies are recommended to help understand (...) the ways in which we can reduce pollution exposure among infants” sounds like taking the easy way out. In the literature, several measures have been proposed to reduce exposure in general and among susceptible populations, including children, and some may have been implemented. Please, elaborate this a little further.

Other +/- major comments:

4. Although familiar with the CCO design, I don't see the use of controlling for daily birth counts. ORs are calculated by comparing the circumstances (air pollution, temperature) of dates of death with these circumstances on control days, so why is the number of births for these days relevant? How could the daily number of births confound the association between death and air pollution?
5. P. 12: “(...) below UK air quality limits.” Please, mention these

limits to allow a proper estimation of the actual pollution level in the study region. Are these the EU limit values? (for now...)

Minor comments

1. Language/typo: p. 4, lines 46-47: "few have ... explored"
2. p. 7: "(...) daily pollution data (...) were calculated by averaging across all monitoring stations with available measurements". Although I know from experience that in a CCO study, it is much more important to have the correct day-to-day variation in air pollution *within* study objects than the geographical variation in exposure *between* subjects, I wonder why you averaged data from all stations instead of attributing data from the nearest station to each subject's residence (or postal code). Would that make any difference? I don't expect so, but readers less familiar to CCO might be happy to read a short statement about your choice (and the ignorable difference with the other approach).
3. P. 8, lines 25-29: According to a theoretical paper by Janes et al. (2005)*, this **time-stratified** selection of control days is indeed the best option to minimize bias. You could refer to this paper to support your choice and mention the term in bold.
4. P. 11, table 2: 1) Units are lacking; 2) the "birth count" column is too narrow for the Birmingham and total numbers.
5. P. 13, table 3: 1) Units are lacking; 2) SIDS and birth count should not have decimals for median, min and max, as these figures are counts; 3) How are the moments for IMD score to be interpreted? Counts, temperatures and pollutants are daily measurements, but what about the time-independent IMD scores?
6. P. 14: Draw the reader's attention to the negative correlations of ozone with other pollutants and to the correlations with temperature: positive (ozone) or negative (others). (here or in the discussion, when explaining the protective effect of ozone).
7. P. 16: "Looking at the temporal distribution (...)" What is the relevance of this paragraph? I think this is only worth mentioning if the air pollution data show the same patterns (years, seasons). Also, table S1 is lacking (for me).
8. P. 20, lines 39-54: By design, season cannot be a confounder in your study, since cases and controls were taken from the same month, and moreover, to control for confounding by temperature within time strata, you inserted temperature as a covariate.

	* Janes H, Sheppard L, Lumley T. Case-crossover analyses of air pollution exposure data: referent selection strategies and their implications for bias. Epidemiology 2005;16:717-26.
--	---

REVIEWER	Chanmin Kim Boston University School of Public Health, USA
REVIEW RETURNED	26-Oct-2017

GENERAL COMMENTS	I have reviewed this manuscript with a particular emphasis on the statistical methods. This manuscript clearly describes their statistical approach. I do have a few comments, however, that I would like the authors to address. 1. For general audiences, could you please explain more about your data structure under the case-crossover design and a conditional Poisson regression? It might be good to spell out the exact equation (with splines) in the appendix. Also, instead of referring R, it would be better for research reproducibility, if you specify which R package and function (and any specification if that was required for implementing the function) you used for the analysis. 2. RE the temperature data, how about using the min. and max. temperature separately instead of using the average of both. I expect that the former can capture more temperature effects than the average.
---

VERSION 1 – AUTHOR RESPONSE

Editorial Requirements:

- Please revise your title to state the research question, study design, and setting (location). This is the preferred format for the journal.

We have amended the title which now reads

"Is ambient air pollution associated with onset of Sudden Infant Death Syndrome? A case crossover study in the UK"

- Please revise the Strengths and Limitations section (after the abstract) to focus on the methodological strengths and limitations of your study rather than summarizing the results.

This has now been rewritten and the text now reads

"Strengths and limitations of this study

- Sudden infant death is the leading cause of death in healthy infants between one month and one year old and our study is the first based on time series data from the UK to investigate the relationship between common air pollutants and SIDS.
- We used a case-crossover and found associations with onset of SIDS and concentration of PM10 and NO2 that lasted days after the initial exposure.
- Though our study is limited in power due to the comparatively small number of daily SIDS events our chosen design is widely applied in air pollution studies and particularly useful for estimating the risk of a rare acute outcome associated with short-term exposure."

Please complete and include a STROBE check-list, ensuring that all points are included and state the page numbers where each item can be found: the check-list can be downloaded from here:

<http://www.strobe-statement.org/?id=available-checklists>

Apologies for the omission, this has now been uploaded.

Comments from the Associate Editor:

I think the research question is of interest (and I didn't find anything about this) but then what I want to know as a physician is whether the authors suggest any public health or clinical implications? The paper makes no mention of it. For example, we know that parental smoking and prone sleeping are the main risk factors for SIDS, which is why there are clear recommendations about not smoking and placing babies "on their back" when sleeping .

This is now discussed in the final paragraph of the Comments section.

Considering confidence intervals for PM10 and NO2 are very wide, the conclusions seem too causal, so I suggest asking them to tone them down.

This is now addressed in the Comments section where the text now reads.

"Though confidence intervals were wide, we found evidence suggesting association of SIDS mortality with PM10 and NO2 exposure."

Reviewer: 1

Reviewer Name: Takashi Yorifuji

Institution and Country: Okayama University, Japan Please state any competing interests: None declared

The present study examined the association between short-term variations in criteria air pollutants and occurrence of SIDS using data from the West Midlands region in the UK. The study is well-conducted and the manuscript is well-written. I have several comments for clarification.

Many thanks for your constructive comments. These we address below.

1. The age range of the participants is not shown. I guess most of the participants are infants. Have now addressed under the section headed "Data on SIDS mortality..." on page 5. The text now reads,

"All cases were between 0 and 12 months old at onset of SIDS"

2. The authors included SIDS mortality for relatively a long period (from 1996 to 2006). I wonder whether the definition of the SIDS has changed during the period. For example, metabolic diseases may have been included in the category of SIDS previously.

As defined by both ICD 9 and 10 metabolic diseases were not included in either and the definition remains the same.

3. It is not slightly clear from the manuscript how the authors assigned air pollution exposure, temperature, or other potential confounders (daily birth counts and IMD) to the participants. Do they only consider temporal variability or do they also consider geographical variability using postcodes? We thank the reviewer for the comment. For each day during the study period, we compiled data on number of SIDS events, average air pollution exposure, minimum and maximum temperature, birth counts and each participant was assigned an IMD category based on their postcode. The analysis is however purely temporal and previously we tried to adjust for IMD as confounder which we have now removed from the analysis. We did not model spatial variability by postcode but do offer descriptive statistics of SIDS rates by post code on page 16. We also present in the supplementary file Figure S1 onset of SIDS in relation to the traffic network.

4. The authors adjusted for daily birth counts and IMD in the model, but I am a little skeptical about this. Because the authors used case-crossover study design, individual unchangeable characteristics like IMD are already adjusted in the design. Moreover, variations in daily birth counts would not affect

occurrence of the SIDS and would not be associated with air pollution, thus daily birth counts does not fulfil the definition of confounding.

We thank the reviewer for pointing this out. We agree with the reviewer and have updated the analysis by excluding IMD and birth counts from the models. We have now further adjusted our model for national holidays instead; please see response to next comment for more details.

5. Rather, some air pollution studies adjust for daily humidity, national holidays, or influenza occurrence which can change daily. These variables should be considered as other potential confounders.

We thank the reviewer for the comment. Some air pollution studies adjust for daily humidity, national holidays, or influenza while many others do not; temperature is one of the most important confounders though and almost all air pollution studies adjust for temperature. We have thus made further adjustment for national holidays by creating an indicator variable. It would have been interesting to control further for influenza and humidity but we were not able to do so as we did not have data for the two variables. Although we believe this would not have substantial impact on our results as in many air pollution studies, we have pointed this out as potential limitation in our study.

6. Discussion is too weak. Some discussions should be added. For example, why did the authors find the positive associations in lag 1 or later lags rather than the lag 0? What is the mechanism of the association between air pollution and SIDS? Other limitations such as potential outcome misclassification or residual confounding should be discussed.

We have re-written the discussion to place our key findings regards PM10, NO2 and O3 in context with the literature. We have also described why we believe it is biologically plausible for there to be a lag with SIDS as observed with other respiratory responses to air pollution. Finally we have addressed the means by which the effects of ambient air pollution might be mitigated at an individual level.

Reviewer: 2

Reviewer Name: Dr. Shereen Hamadneh

Institution and Country: Al-albait University, Jordan Please state any competing interests: None declared

This study aimed to examine the relationship between short-term variations in criteria air pollutants and occurrence of SIDS, would you please include research question or study objective and clearly defined in the introduction section under the problem statement. Good luck

Many thanks for your positive response. We have now updated the introduction to clarify the research question and the objectives of the study and included the line at the end of the first paragraph.

“...and here we examine the effects of the short term variations in air pollution and the onset of SIDS.”

Reviewer: 3

Reviewer Name: Hans Scheers

Institution and Country: currently: P95 Koning, Leopold III, laan 1 3001, Heverlee Belgium
formerly: KU Leuven, Department of Public Health, Herestraat 49 3000, Leuven Belgium Please state any competing interests: none declared

Comments for authors

In this paper, the authors investigated the association between Sudden Infant Death Syndrome (SIDS) and atmospheric air pollution in a densely populated region in the United Kingdom. The

authors applied a case-crossover (CCO) design, a widely applied method to study acute health effects in association with short-term changes in air pollution.

The authors found a significantly higher risk for SIDS with increasing levels of PM₁₀ and NO₂. For other pollutants, however, no such associations were detected, and O₃ had a protective effect.

The authors conclude that air pollution exposure may increase risk of SIDS mortality.

The paper attracts attention by its conciseness, fluent language and clear message. However, the brevity of the manuscript has some drawbacks. Possible questions raised by the results remain unanswered in the discussion, and I also have a couple of questions on the methodology.

Many thanks for your constructive comments the time you have taken to produce these is much appreciated and these we address in turn below.

My detailed comments are as follows:

Major comments

My major comment concerns the brevity of the “Comments” section and subsequent lack of in-depth discussion. I see at least three points that deserve more attention:

We have now rewritten the Comments section.

1. Finding stronger associations with PM₁₀ and NO₂ than with other pollutants is in line with many other studies that included several pollutants. Particularly PM is considered the most harmful of all pollutants. You may want to mention this accordance and refer to some studies.

We now address this under the heading “Detailed discussion”. The text now reads

“The associations we found with PM₁₀ and NO₂ are in line with those of previous studies. Woodruff and Lipfert found a significantly increased risk of SIDS per 10 µg/m³ increase in PM₁₀. [6, 38] The association we describe between PM₁₀ and infant mortality at concentrations below EU guidelines (i.e., 50 µg/m³) has also been reported previously [Rice et al 2013; Samoli 2013; Scheers 2013] amidst the realisation that particulate matter is the most significant contributor to air quality related morbidity and mortality [Lancet commission on air quality and health 2017].

“Certain groups of the population are more vulnerable to ambient air pollution than others and children figure predominantly amongst them due to the fragility of their immune system and the ratio of their lung capacity to their size [Pope 2006]. Particulate matter has been recognised as the most significant contributor to air quality related morbidity and mortality [Lancet commission on air quality and health Published online October 19, 2017 [http://dx.doi.org/10.1016/S0140-6736\(17\)32345-0](http://dx.doi.org/10.1016/S0140-6736(17)32345-0)] and a number of studies have reported a link between infant mortality and PM₁₀ at similar concentrations of pollutant below air quality guidelines as are described in this study [Rice et al 2013; Samoli 2013; Scheers 2013; Yorifuji et al 2017]. However previous evidence of a direct association with SIDS is inconclusive and few studies describe a significant association of PM₁₀ with increased risk [6, 38]. “

References

Lancet commission on air quality and health Published online October 19, 2017
[http://dx.doi.org/10.1016/S0140-6736\(17\)32345-0](http://dx.doi.org/10.1016/S0140-6736(17)32345-0)

Rice MB, Ljungman PL, Wilker EH, et al. Short-term exposure to air pollution and lung function in the Framingham Heart Study. *Am J Respir Crit Care Med* 2013; 188 (11): 1351- 357.

Samoli E, Stafoggia M, Rodopoulou S, et al. Associations between fine and coarse particles and mortality in Mediterranean cities: results from the MED-PARTICLES project. *Environ Health Perspect* 2013; 121 (8):932–938]

Hans Scheers H, Mwalili SM, Faes C, Fierens F, Nemery B, and Nawrot TS. Does Air Pollution Trigger Infant Mortality in Western Europe? A Case Crossover Study *Environ Health Perspect* 2011; 119:1017–1022.

2. Finding stronger delayed effects (lags 3 to 6) than immediate effects (lags 0,1,2), however, is not in line with many CCO studies on air pollution. Please discuss probable mechanisms that might explain a delayed effect for SIDS mortality, in contrast to the immediate effects usually found when studying other outcomes, such as myocardial infarct or overall mortality.

The text now reads

“The issue of lag is interesting we found the strongest association with SIDS with PM10 and NO2 occurred over several lags in line with previous studies that have found evidence of a lag between exposure to criteria pollutants and mortality. For PM10, a single days exposure has been shown to have an effect for up to 5 days [Schwartz 2000] [Massimo 2013] and numbers of respiratory deaths appear to be more affected by air pollution levels on previous days, than cardiovascular deaths that are impacted by same-day pollution. [Braga et al 2001]. For example myocardial infarction represents an acute response to a trigger [Gold 2000].

This is biologically plausible when considering how a wider and more lagged response can be expected for deaths from respiratory responses to pollution (eg, COPD) via the proposed mechanisms, such as pulmonary or systemic inflammation and modulated immunity [Blackwell et al 2015].”

References

Schwartz J The distributed lag between air pollution and daily deaths *Epidemiology*. 2000 May;11(3):320-6.

Association Between Short-Term Exposure to PM2.5 and PM10 and Mortality in Susceptible Subgroups: A Multisite Case-Crossover Analysis of Individual Effect Modifiers Rita E Massimo A, Annunziata S, Giovanna F, Canova BC, De Togni A, Di Biagio K, Gherardi B, Giannini S, Lauriola P *American Journal of Epidemiology*, Volume 184, Issue 10, 15 November 2016, Pages 744–754, <https://doi.org/10.1093/aje/kww078>

Braga AL, Zanobetti A, Schwartz J. The lag structure between particulate air pollution and respiratory and cardiovascular deaths in 10 US cities. *J Occup Environ Med*. 2001 Nov;43(11):927-33.

Gold DR, Litonjua A, Schwartz J, et al. Ambient pollution and heart rate variability. *Circulation*. 2000; 101: 1267–1273

Blackwell C, Moscovis S, Hall S, et al. Exploring the risk factors for sudden infant deaths and their role in inflammatory responses to infection. *Front Immunol*. 2015;6:44.

Also, the figure presenting results adjusted for IMD (Figure S2) shows remarkably high ORs for lag 6, compared to the other lag structures for most pollutants. Are you sure this figure is correct? It is hard to believe that adjusting for IMD leads to such a different figure than the one showing crude associations (Figure 2).

Our results indicated a delayed effect of air pollution (lags 1-6 but less so for lag 0). That is, Air pollution impacts can be felt at least up to six days after exposure. We have now added literature to support and explain this observed delayed effect.

We have also excluded IMD from the model as confounder based on comments from reviewers and have updated all figures. We agree that IMD is unlikely to confound the SIDS-air pollution relationship in this CCO design.

3. Next to biological mechanisms, I also would like to read some more discussion on the public health impact of your study. Just writing “Thus, future studies are recommended to help understand (...) the ways in which we can reduce pollution exposure among infants” sounds like taking the easy way out. In the literature, several measures have been proposed to reduce exposure in general and among susceptible populations, including children, and some may have been implemented. Please, elaborate this a little further.

The text now reads

Concern around the effects of air pollution continue to mount particularly within developing economies (WHO 2014) where it contributes to 3.3 million premature deaths worldwide per year a figure estimated to double by 2050 if the issue remains unattended (Lelieveld et al, 2015). However it appears that recently fossil fuel emissions have begun to increase again due to the developing world's reliance on them to power their expanding economies [LeQuere 2017]. In the developed West, policy initiatives are beginning to recognise the mounting issue posed by the adverse effects on health posed by ambient air pollution. However shifting political priorities in the United States has seen a commitment to revitalise the coal industry [<http://www.independent.co.uk/news/world/americas/donald-trump-coal-mining-jobs-promise-experts-disagree-executive-order-a7656486.html>] and an increase in the production of shale oil [<https://www.eia.gov/tools/faqs/faq.php?id=847&t=6>] However mitigating these risks is not a straightforward proposition and government policies appear slow to react, for example in the UK pledges to cease sales of diesel and petrol cars do not come into effect until 2040 [Guardian 2017]. In the absence of coherent policy to address the issue a number of proposals have been put forward to mitigate the effects of high levels of ambient exposure. Knowing that children and young adults may be highly susceptible to some of the subclinical changes caused by air pollution (Rich et al 2012; Wright et al 2013) and as indoor concentrations are lower than ambient levels advice has been to remain indoors to reduce exposure and acute health risks on high air pollution days (Plaia et al 2011). There are also systems available for cleaning indoor air though these may be deemed expensive for the economically deprived (Macintosh 2008). There has also been a case made for chemo-preventive interventions, such as antioxidant or antithrombotic agents, but without data on health outcomes, no recommendations can be made in their use for primary prevention (Laumbach et al 2015).

References

- WHO (2014) “Burden of Disease from Ambient Air Pollution for 2012.” n.d. (http://www.who.int/phe/health_topics/outdoorair/databases/AAP_BoD_results_March2014.pdf)
- Lelieveld, J., J. S. Evans, M. Fnais, D. Giannadaki, and A. Pozzer. 2015. “The Contribution of Outdoor Air Pollution Sources to Premature Mortality on a Global Scale.” *Nature* 525 (7569). [nature.com](http://www.nature.com): 367–71.
- Earth System Science Data Discussions, DOI: 10.5194/essdd-2017-123; *Nature Climate Change*, DOI: 10.1038/s41558-017-0013-9; *Environmental Research Letters*, DOI: 10.1088/1748-9326/aa9662
- Plaia A, Ruggieri M: Air quality indices: a review. *Rev Environ Sci Bio* 2011;10:165-79
- Rich DQ, Kipen HM, Huang W, et al. Association between changes in air pollution levels during the Beijing Olympics and biomarkers of inflammation and thrombosis in healthy young adults. *JAMA* 2012;307:2068-78.
- Brunst KJ. Programming of respiratory health in childhood: influence of outdoor air pollution. *Curr Opin Pediatr* 2013;25:232-9.
- Wright RJ, Brunst KJ. Programming of respiratory health in childhood: influence of outdoor air pollution. *Curr Opin Pediatr* 2013;25:232-9.
- Laumbach R, Meng Q, Kipen H. What can individuals do to reduce personal health risks from air pollution? *J Thorac Dis* 2015;7(1):96-107. doi: 10.3978/j.issn.2072-1439.2014.12.2 <http://www.independent.co.uk/news/world/americas/donald-trump-coal-mining-jobs-promise-experts-disagree-executive-order-a7656486.html>

<https://www.eia.gov/tools/faqs/faq.php?id=847&t=6>

David L. MacIntosh , Theodore A. Myatt , Jerry F. Ludwig , Brian J. Baker , Helen H. Suh & John D. Spengler (2008) Whole House Particle Removal and Clean Air Delivery Rates for In-Duct and Portable Ventilation Systems, Journal of the Air & Waste Management Association, 58:11, 1474-1482
Other +- major comments:

4. Although familiar with the CCO design, I don't see the use of controlling for daily birth counts. ORs are calculated by comparing the circumstances (air pollution, temperature) of dates of death with these circumstances on control days, so why is the number of births for these days relevant? How could the daily number of births confound the association between death and air pollution?
We thank the reviewer for pointing this out. We agree with the reviewer and have updated the analysis by excluding birth counts from the models.

5. P. 12: "(...) below UK air quality limits." Please, mention these limits to allow a proper estimation of the actual pollution level in the study region. Are these the EU limit values? (for now...)

Now clarified and the text reads,

"The daily average air pollution concentrations and their standard deviations are presented in Table 3 and show that average concentrations tended to be below UK air quality limits as defined by the EU Ambient Air Quality Directive [Defra UK EU limits]."

Reference

<https://uk-air.defra.gov.uk/air-pollution/uk-eu-limits>

Minor comments

1. Language/typo: p. 4, lines 46-47: "few have ... explored"

Thank you now amended.

2. p. 7: "(...) daily pollution data (...) were calculated by averaging across all monitoring stations with available measurements". Although I know from experience that in a CCO study, it is much more important to have the correct day-to-day variation in air pollution within study objects than the geographical variation in exposure between subjects, I wonder why you averaged data from all stations instead of attributing data from the nearest station to each subject's residence (or postal code). Would that make any difference? I don't expect so, but readers less familiar to CCO might be happy to read a short statement about your choice (and the ignorable difference with the other approach).

We thank the reviewer for the comment. We have now updated the statement ""(...) daily pollution data (...) were calculated by averaging across all monitoring stations with available measurements as our models were based on the temporal relationship between air pollution and SIDS and did not fit a spatial model." in the methods section.

3. P. 8, lines 25-29: According to a theoretical paper by Janes et al. (2005)*, this time-stratified selection of control days is indeed the best option to minimize bias. You could refer to this paper to support your choice and mention the term in bold.

* Janes H, Sheppard L, Lumley T. Case-crossover analyses of air pollution exposure data: referent selection strategies and their implications for bias. *Epidemiology* 2005;16:717-26.

Thank you - this reference has now been cited and the phrase "time stratified" explicitly used.

"We applied the time stratified case-crossover approach where the strata are matching days based on the same day of the week, calendar month and year..."

4. P. 11, table 2: 1) Units are lacking; 2) the “birth count” column is too narrow for the Birmingham and total numbers.

The table has now been altered and the units inserted.

5. P. 13, table 3:

1) Units are lacking;

Now included

2) SIDS and birth count should not have decimals for median, min and max, as these figures are counts;

Now amended

3) How are the moments for IMD score to be interpreted? Counts, temperatures and pollutants are daily measurements, but what about the time-independent IMD scores?

IMD scores are time independent but based on the participant’s residence; that is, summary based on all participants irrespective of time.

6. P. 14: Draw the reader’s attention to the negative correlations of ozone with other pollutants and to the correlations with temperature: positive (ozone) or negative (others). (here or in the discussion, when explaining the protective effect of ozone.

Thank you, have now added the following text on page 14.

“With regards the apparent protective properties of O₃ that we describe here it’s worth noting that these have been seen previously [33]. One possible reason for the negative association of O₃ and SIDS might be its negative correlation with other pollutants, in particular that with fine particulate matter pollution [40].”

7. P. 16: “Looking at the temporal distribution (...)” What is the relevance of this paragraph? I think this is only worth mentioning if the air pollution data show the same patterns (years, seasons). Also, table S1 is lacking (for me).

We thank the reviewer for the comment. We have accepted the reviewer’s suggestion and removed the sentence mentioned “Looking at the temporal distribution (...)”.

We have also removed the reference to table S1.

8. P. 20, lines 39-54: By design, season cannot be a confounder in your study, since cases and controls were taken from the same month, and moreover, to control for confounding by temperature within time strata, you inserted temperature as a covariate.

We thank the reviewer for the comment. We have now withdrawn the reference to season as potential confounder/effect modifier. We have instead referred to the negative correlation with temperature.

* References: Six studies (refs. 5-10) and two reviews (refs. 16-17) on the association between SIDS and air pollution are mentioned. The most recent one (ref. 10) dates from 2007. I would think that more recent references can be found.

Thank you additional references have now been included;

Yorifuji T, Kashima S, Doi H. Acute exposure to fine and coarse particulate matter and infant mortality in Tokyo, Japan (2002-2013). *Sci Total Environ.* 2016 May 1;551-552:66-72.

doi:10.1016/j.scitotenv.2016.01.211. Epub 2016 Feb 11.

Luechinger S Air pollution and infant mortality: A natural experiment from power plant desulfurization *Journal of Health Economics* 37 (2014) 219–231

Goldwater PN Infection: the neglected paradigm in SIDS research *Archives of Disease in Childhood* 2017;102:767-772.

Reviewer: 4

Reviewer Name: Chanmin Kim

Institution and Country: Boston University School of Public Health, USA Please state any competing interests: None declared

I have reviewed this manuscript with a particular emphasis on the statistical methods. This manuscript clearly describes their statistical approach. I do have a few comments, however, that I would like the authors to address.

1. For general audiences, could you please explain more about your data structure under the case-crossover design and a conditional Poisson regression? It might be good to spell out the exact equation (with splines) in the appendix. Also, instead of referring R, it would be better for research reproducibility, if you specify which R package and function (and any specification if that was required for implementing the function) you used for the analysis.

We thank the reviewer for the comment. We have now added more description detailing the packages used with necessary data preparations stages and providing sample line of R code to fit the conditional Poisson model using the R package 'gnm' in the appendix as recommended. The methods are adapted from the paper by Armstrong et al. 2014 where more details including model specification can be found. i.e.

```
CPR_model <- gnm(sids_freq ~ Lag(pollution, k) + ns(temperature,3) + holiday, data=data, family=quasipoisson, eliminate=factor(stratum))
```

Where,

Lag(pollution, k): pollution exposure at lag k

ns(temperature,3): temperature spline

sids_freq: SIDS count

stratum: stratum indicator (for yearXmonthXday-of-week)

2. RE the temperature data, how about using the min. and max. temperature separately instead of using the average of both. I expect that the former can capture more

We thank the reviewer for the comment. We have now added results from a sensitivity analysis using minimum temperature instead of the average and presented the results in supplementary Figure S2.

VERSION 2 – REVIEW

REVIEWER	Chanmin Kim Boston University, USA
REVIEW RETURNED	26-Dec-2017

GENERAL COMMENTS	The paper is clearly written and all previous concerns are properly taken care of except a missing supplementary material about the R packages, sample code for a conditional Poisson regression model, etc. On page 10, the authors state that these are given in the supp. material, but I couldn't locate the file. Without this information, the study cannot be easily repeated. minor : Ref. # 37 and #39 are redundant.
--

REVIEWER	Takashi Yorifuji Okayama University, Japan
-----------------	---

REVIEW RETURNED	03-Jan-2018
GENERAL COMMENTS	The authors responded to my concerns and I have no additional comment.
REVIEWER	Hans Scheers P95 Koning Leopold III laan 1 3001 Heverlee Belgium
REVIEW RETURNED	15-Jan-2018
GENERAL COMMENTS	All my questions and remarks have been answered adequately, thank you. The discussion is now more interesting to read and much more convincing concerning the societal importance of the study results. The following list contains a handful of final cosmetic changes I would like to suggest:  * page 5, line 15. Start new sentence: "Here, we examine ..." * page 5, line 40. SIDS (all capital letters) * page 9, lines 31-38: "same day of the week, calendar month and year" is repeated in two consecutive sentences. I would keep it in the second one only, so the first would read: "We applied the time-stratified case-crossover approach, which has previously been used to minimise bias." What time-stratified exactly means, is explained in the next sentence. page 20, lines 38-45. This sentence sounds quite odd. Please rephrase. page 21, line 20: "natively" should be "negatively" page 27-28: references 34 and 46 are duplicates.

VERSION 2 – AUTHOR RESPONSE

Editorial Requirements:

- Please tone down the conclusions presented in the abstract, as per the previous comments from the Associate Editor.

Thank you conclusion now reads,

“Conclusions: The results indicated ambient air pollutants, particularly PM10 and NO2, may show some association with increased SIDS mortality. Thus, future studies are recommended to understand possible mechanistic explanations on the role of air pollution on SIDS incidence and the ways in which we might reduce pollution exposure among infants.”

- Unfortunately, the STROBE checklist provided is incomplete. Please complete and include a full STROBE check-list, ensuring that all points are included and state the page numbers where each item can be found. The full check-list can be downloaded from here: <http://www.strobe-statement.org/?id=available-checklists>

Apologies as you know there is no checklist specifically designed for case-crossover studies so have used the checklist for case-control studies which is the closest to it. We are happy to take your advice if there is another version that you would like us to use.

Reviewer(s)' Comments to Author:

Reviewer: 1

Reviewer Name: Takashi Yorifuji

Institution and Country: Okayama University, Japan Please state any competing interests: None declared.

Please leave your comments for the authors below The authors responded to my concerns and I have no additional comment.

Thank you.

Reviewer: 3

Reviewer Name: Hans Scheers

Institution and Country: P95, Koning Leopold III laan 1, 3001 Heverlee, Belgium Please state any competing interests: none declared

Please leave your comments for the authors below All my questions and remarks have been answered adequately, thank you. The discussion is now more interesting to read and much more convincing concerning the societal importance of the study results.

The following list contains a handful of final cosmetic changes I would like to suggest:

* page 5, line 15. Start new sentence: "Here, we examine ..."

Now added

* page 5, line 40. SIDS (all capital letters)

Amended

* page 9, lines 31-38: "same day of the week, calendar month and year" is repeated in two consecutive sentences. I would keep it in the second one only, so the first would read: "We applied the time-stratified case-crossover approach, which has previously been used to minimise bias." What time-stratified exactly means, is explained in the next sentence.

Thank you, now amended

page 20, lines 38-45. This sentence sounds quite odd. Please rephrase.

Now rephrased

"The potential association with NO₂ we discovered reflected the findings of a multi-city study from Canada that found an increased risk for SIDS with a lag of two-days.[8] More broadly our findings are in line with previous studies that have found evidence of a lag between exposure to criteria pollutants and mortality.

page 21, line 20: "natively" should be "negatively"

Thank you now amended

page 27-28: references 34 and 46 are duplicates.

We have updated the bibliography.

Reviewer: 4

Reviewer Name: Chanmin Kim

Institution and Country: Boston University, USA Please state any competing interests: None declared

Please leave your comments for the authors below

The paper is clearly written and all previous concerns are properly taken care of except a missing supplementary material about the R packages, sample code for a conditional Poisson regression model, etc. On page 10, the authors state that these are given in the supp. material, but I couldn't locate the file. Without this information, the study cannot be easily repeated.

Apologies an additional Supplementary File (S3) has now been uploaded.

minor : Ref. # 37 and #39 are redundant.

We have now removed the duplicate reference and updated the bibliography.

VERSION 3 – REVIEW

REVIEWER	Chanmin Kim Biostatistics, Boston University
REVIEW RETURNED	18-Feb-2018
GENERAL COMMENTS	All my previous concerns have been fully addressed. Typo : page 35 line 24: temprture -> temperature